# Biocompatibility and Biological Performance Evaluation of Additive-Manufactured Bioabsorbable Iron-Based Porous Suture Anchor in a Rabbit Model

**DOI:** 10.3390/ijms22147368

**Published:** 2021-07-08

**Authors:** Chien-Cheng Tai, Hon-Lok Lo, Chen-Kun Liaw, Yu-Min Huang, Yen-Hua Huang, Kuo-Yi Yang, Chih-Chieh Huang, Shin-I Huang, Hsin-Hsin Shen, Tzu-Hung Lin, Chun-Kuan Lu, Wen-Chih Liu, Jui-Sheng Sun, Pei-I Tsai, Chih-Yu Chen

**Affiliations:** 1Internal Ph.D. Program for Cell Therapy and Regeneration Medicine, College of Medicine, Taipei Medical University, Taipei 11031, Taiwan; d151107007@tmu.edu.tw (C.-C.T.); rita1204@tmu.edu.tw (Y.-H.H.); 2Department of Orthopedic Surgery, Kaohsiung Medical University Hospital, Kaohsiung Medical University, Kaohsiung 80756, Taiwan; honlok1021@hotmail.com (H.-L.L.); andysirliu@gmail.com (W.-C.L.); 3Department of Orthopedics, School of Medicine, College of Medicine, Taipei Medical University, Taipei 11031, Taiwan; 17255@s.tmu.edu.tw (C.-K.L.); yellowcorn0326@yahoo.com.hk (Y.-M.H.); 4Department of Orthopedics, Shuang Ho Hospital, Taipei Medical University, No. 291, Zhongzheng Rd., Zhong He Dist., New Taipei City 23561, Taiwan; 5Research Center of Biomedical Device, Graduate Institute of Biomedical Optomechatronics, College of Biomedical Engineering, Taipei Medical University, Taipei 11301, Taiwan; 6Department of Biochemistry and Molecular Cell Biology, School of Medicine, College of Medicine, Taipei Medical University, Taipei 11031, Taiwan; 7TMU Research Center of Cell Therapy and Regeneration Medicine, Taipei Medical University, Taipei 11031, Taiwan; 8Biomedical Technology and Device Research Laboratories, Industrial Technology Research Institute, Chutung, Hsinchu 310401, Taiwan; yangkuoyi@itri.org.tw (K.-Y.Y.); Sigher@itri.org.tw (C.-C.H.); sophiashini@itri.org.tw (S.-IH.); shenhsin@itri.org.tw (H.-H.S.); peiyi@itri.org.tw (P.-IT.); 9Material and Chemical Research Laboratories, Industrial Technology Research Institute, Hsinchu 31040, Taiwan; DustyLin@itri.org.tw; 10Department of Orthopaedics, Park One International Hospital, Kaohsiung 813322, Taiwan; u9001054@yahoo.com.tw; 11Ph.D. Program in Biomedical Engineering, College of Medicine, Kaohsiung Medical University, Kaohsiung 807378, Taiwan; 12Regeneration Medicine and Cell Therapy Research Center, Kaohsiung Medical University, Kaohsiung 807378, Taiwan; 13Department of Orthopedics, China Medical University, Taichung 40202, Taiwan; drjssun@cmu.edu.tw; 14School of Medicine, College of Medicine, China Medical University, Taichung 40202, Taiwan

**Keywords:** additive manufacturing (3D printing), bioabsorbable, iron-based, suture anchor

## Abstract

This study evaluated the biocompatibility and biological performance of novel additive-manufactured bioabsorbable iron-based porous suture anchors (iron_SAs). Two types of bioabsorbable iron_SAs, with double- and triple-helical structures (iron_SA_2_helix and iron_SA_3_helix, respectively), were compared with the synthetic polymer-based bioabsorbable suture anchor (polymer_SAs). An in vitro mechanical test, MTT assay, and scanning electron microscope (SEM) analysis were performed. An in vivo animal study was also performed. The three types of suture anchors were randomly implanted in the outer cortex of the lateral femoral condyle. The ultimate in vitro pullout strength of the iron_SA_3_helix group was significantly higher than the iron_SA_2_helix and polymer_SA groups. The MTT assay findings demonstrated no significant cytotoxicity, and the SEM analysis showed cells attachment on implant surface. The ultimate failure load of the iron_SA_3_helix group was significantly higher than that of the polymer_SA group. The micro-CT analysis indicated the iron_SA_3_helix group showed a higher bone volume fraction (BV/TV) after surgery. Moreover, both iron SAs underwent degradation with time. Iron_SAs with triple-helical threads and a porous structure demonstrated better mechanical strength and high biocompatibility after short-term implantation. The combined advantages of the mechanical superiority of the iron metal and the possibility of absorption after implantation make the iron_SA a suitable candidate for further development.

## 1. Introduction

Inert metals such as stainless steel, titanium alloys, and cobalt alloys have been widely used as orthopedic and cardiovascular implants because they demonstrate excellent corrosion resistance and possess adequate mechanical properties relative to local biological tissues [1,2,3]. Conventionally, these inert metallic implants are designed to permanently remain in the body until interventional removal [3,4]. However, the use of permanent inert materials for providing temporary support can cause several complications, such as stress shielding over time, leading to the weakening of the implanted tissue, development of foreign body sensations, distortion of diagnostic images, and requirement of secondary surgery to remove the implants [1,4,5]. Therefore, the use of self-degradable metallic implants in the body environment and the gradual transfer of the load onto the healing tissue until tissue recovery can be effective strategies for overcoming the drawbacks of inert metallic implants.

The majority of currently available bioabsorbable implants used clinically are made from biodegradable synthetic polymer compounds and have been approved by the Food and Drug Administration (FDA) [4,6,7]. However, these implants, prepared using biodegradable synthetic polymers, are usually not adequately stiff to be used in major load-bearing applications [8,9]. By contrast, recent advances in the development of biodegradable metallic materials have demonstrated the potential to revolutionize metallic implant designs and treatment strategies [4,5,7,10,11,12]. Implants prepared using biodegradable metals are significantly stronger than those developed using polymers; moreover, degraded metal particles are fully dissolvable in body fluids [5].

Iron-based biodegradable materials are considered to be a suitable candidate for use as metallic implants [13,14,15,16]. These materials have been demonstrated to have satisfactory cytocompatibility, and their mechanical properties can match those of the natural bone [13,14,17,18]. However, the slow degradation of iron-based biodegradable materials in the physiological environment is a major weakness, and this drawback should be overcome to enable their use in clinical applications [5,13,14,19].

Increasing surface area and incorporation of porous structures into an iron-based biodegradable material can accelerate its degradation rate [5,20,21,22,23]. Our control group suture anchors were famous for its open architecture in appearance that had the higher surface area [24,25]. In this study, we innovated bioabsorbable iron-based porous suture anchors (hereafter referred to as iron SAs) by using control group suture anchor as reference template. On the other hand, in order to compare biomechanical performances among different appearances, we constructed our suture anchors with two different appearances (a double- and triple-helical structure, respectively) and incorporated porous structures in the threads by using additive manufacturing (AM) technology. We evaluated the biocompatibility (both in vivo and in vitro), biomechanical performance, micro-computed tomography (micro-CT) results, and histopathological analysis findings of iron SAs and compared them with those of synthetic polymer-based suture anchors (hereafter referred to as polymer SAs) by using a rabbit animal model. We hypothesized that iron SAs produced using AM technology are biocompatible and can outperform currently used polymer SAs.

## 2. Results

### 2.1. In Vitro Mechanical Analyses of Bioabsorbable Iron SA

The specifications and geometrical appearances of the two types of suture anchors are shown in Figure 1A,B.

The ultimate in vitro pullout strength was significantly higher for the iron SA_3 helix (270.94 ± 34.76 N) than for the iron SA_2 helix (218.43 ± 27.53 N, *p* = 0.0158) and the polymer SA (168.54 ± 29.45 N, *p* = 0.0003). In addition, the ultimate in vitro pullout strength was significantly higher for the iron SA_2 helix than for the polymer SA (*p* = 0.0126). These results are shown in Figure 1B and Table 1.

### 2.2. In Vitro Biocompatibility Analyses of Bioabsorbable Iron SA

As shown in Figure 2A and Table 2, the findings of the MTT assay demonstrated no significant cytotoxicity in the two groups treated with the iron SA extracts. In addition, the results of the SEM analysis exhibited the attachment of cells onto the implant surface accompanied with lamellipodial and filopodial extrusions from the cells (Figure 2B). These findings suggest that cells could attach onto the implant surface and explore the environment without experiencing significant cytotoxicity.

### 2.3. In Vivo Biomechanical Analysis

The findings of biomechanical analysis revealed that the ultimate failure load of the iron SA_3 helix group at 4 weeks (162.36 ± 23.02 N) was significantly stronger than that of the iron SA_2 helix group (216.23 ± 20.07 N, *p* = 0.0004) and the polymer SA group (257.85 ± 38.66 N, *p* = 0.0413). At 12 weeks, the ultimate failure load of the iron SA_2 helix and iron SA_3 helix groups (318.59 ± 36.58 N and 361.97 ± 69.45 N, respectively) were significantly higher than that of the polymer SA group (198.80 ± 14.28 N; *p* = 0.0001 and *p* = 0.0002, respectively). However, at 12 weeks, the ultimate pullout strength did not significantly differ between the iron SA_2 helix and iron SA_3 helix groups (*p* = 0.2056). The data are shown in Figure 3 and Table 3.

The ultimate failure load significantly increased from 4 to 12 weeks in all the groups. The results are shown in Table 3.

### 2.4. Micro-CT Analysis

Micro-CT was performed to evaluate bone formation between the implant and bone tissue (Figure 4A). Compared with the polymer SA group, the iron SA groups had higher BV/TV at both 4 and 12 weeks postoperatively; however, this difference was not significant (*p* = 0.8102 and *p* = 0.2395 at 4 weeks and *p* = 0.5319 and *p* = 0.4097 at 12 weeks, respectively). The results are shown in Figure 4B and Table 4. These results suggest that both the polymer SA and iron SA resulted in favorable bone growth. In addition, the iron SA groups had a higher BS/TV percentage 12 weeks postoperatively (Figure 4C and Table 4); however, this difference was not statistically significant (*p* = 0.1656 and *p* = 0.1630, respectively). A higher BS/TV indicated increased bone growth closer (<1000 μm) to the implant surface region. Our results demonstrated that compared with the polymer SA, the iron SA led to increased bone growth, especially at the region near the implant surface.

The results of iron SA degradation analysis showed that the ST decreased and the SSV/OV increased sequentially from the preoperative period until 12 weeks postoperatively (Figure 5A,B and Table 5, respectively). These findings are also illustrated in the reconstructed micro-CT images shown in Figure 5C. These results indicate that the iron SA gradually degraded into smaller fragments, had decreased structural thickness, and had an increased total surface area after implantation.

### 2.5. Biochemical Analysis

For the biochemical analysis, blood samples were collected from all the rabbits immediately preoperatively and 4, 8, and 12 weeks postoperatively. The concentrations of iron, ALT, Cr, and BUN were determined. The results are shown in Figure 6 and Table 6.

### 2.6. Histological and Histopathological Analyses

Bone formation was observed both in the polymer SA and iron SA groups (Figure 7). In all histological analyses, particularly at a high magnification field, new bone growth was observed in the region closely contacted to the suture anchors. The findings of the polymer SA are compatible with those of previous studies [26,27]. Our results revealed high biocompatibility of the iron SA; this finding is compatible with those of the MTT assay and SEM analysis (Figure 2A,B).

The histopathological findings of the liver, spleen, heart, and kidneys did not significantly differ between the iron SA group and the polymer SA group. Figure 8 shows the histopathological specimens of the visceral organs obtained from one of the animals in the iron SA_3 helix group. Figure 9 presents the findings of the Prussian blue staining of the liver and spleen, and Table 7 lists the semi-quantitative results.

## 3. Discussion

Our iron SAs had double- or triple-helical threads, a porous structure, and an open architecture design to maximize the contact surface, increase initial stability, and facilitate bone growth (Figure 1A). The findings of the in vitro mechanical analysis revealed that the ultimate pullout strength of the iron SA group was higher than that of the polymer SA group (Figure 1B and Table 1). Poly(lactic-co-glycolic acid) is the main material (accounting for 65%) of a polymer SA [28,29]. However, our iron SA was prepared using metals; thus, the mechanical strength of the iron SA was higher than that of the polymer SA. Higher mechanical strength contributed to the higher pullout strength of the iron SA groups. In addition, the triple-helical design showed more favorable mechanical performance compared with the double-helical design. These findings suggest that not only a higher mechanical strength but also a larger contact surface area resulted in more satisfactory ultimate pullout performance. On the other hand, these results also demonstrated the significantly better implant-bone integration in the iron-based SA groups at 12 weeks after implantation compared to those at 4 weeks after implantation.

Biocompatibility is a major concern for an iron-based bioabsorbable material. A biocompatible material should exert minimal inflammatory and toxic effects both locally and systemically after its degradation [5,30,31,32]. In addition, the biocompatibility of a product’s surface is crucial [30]. Our iron-based SAs demonstrated satisfactory biocompatibility from several aspects. First, the results of the MTT assay and SEM analysis demonstrated no significant cytotoxicity of the extract or implant surface (Figure 2 and Table 2). Second, the in vivo biochemical analysis (Figure 6 and Table 6) and histopathological (Figure 8) analysis of the visceral organs revealed no significant increase in iron, ALT, Cr, and BUN concentrations and tissue toxicity. Moreover, semi-quantitative results for iron ion staining in the liver and spleen did not significantly differ between the iron SA and polymer SA groups (Figure 9 and Table 7). These findings are compatible with those of previous studies reporting the biocompatibility of iron-based implants [4,5,31,33,34]. In summary, iron SAs were biocompatible after short-term implantation in a rabbit animal model.

The synthetic polymer SA used in this study consisted of 20% calcium sulfate and 15% β-TCP [27,28,29] and was expected to result in favorable bone growth. Our histological and micro-CT examination demonstrated favorable bone growth; these findings are compatible with those of previous studies [27,28,29]. As shown in Figure 2B, the results of SEM analysis demonstrated cell attachment on the implant with lamellipodial and filopodial extrusions; this result is compatible with the histopathological finding (Figure 7) that demonstrated satisfactory cell contact with the implant. In addition, the micro-CT results revealed similar BV/TV and BS/TV between the polymer SA and iron SA groups (Figure 4 and Table 4), suggesting that all implants showed similar bone growth. These findings suggest that both the polymer and iron SAs resulted in similar total bone growth and demonstrated bone formation closer to the implant surface.

Similar to a polymer SA that shows favorable contact interface biocompatibility, iron SAs demonstrated bone–implant contact interface biocompatibility, as observed in SEM and histological analyses (Figure 2 and Figure 7). This finding is compatible with those of previous studies [5,13,30].

According to previous literatures, metallic SA had superior pullout strength compared to polymer SA [35,36,37]. In addition, increased threads surface area of SA (Figure 1B) also contributed to better pullout strength [38,39]. In combination of material superiority, triple-helical geometry design to increase thread-bone contact surface area, and integrated porous design of iron SAs could maximize the bone–implant contact interface and increase its biocompatibility and result in better in vivo biomechanical pullout strength of the iron SA than that of the polymer SA (Figure 3 and Table 3).

To more effectively delineate and quantify the implant degradation profile, we performed micro-CT analysis. As shown in Figure 5 and Table 5, the implant structure thickness and SSV percentage at 4 and 12 weeks postoperatively significantly decreased after implantation compared with before implantation. However, the implant structure thickness and SSV did not significantly differ between 4 and 12 weeks after surgery. These findings indicate that the iron implant rapidly degraded during the first 4 weeks after implantation and then slowly degraded after 4 weeks. This finding is compatible with those of previous studies indicating that the initial oxidation of iron formed a protective layer, thus preventing further degradation [5,13,30,33,34].

Our study has some limitations that should be addressed. First, the study was only conducted for 12 weeks. Although our short-term in vivo biochemical and histopathological analyses showed high biocompatibility of iron SAs, future clinical studies should examine their long-term biocompatibility. Second, local and systemic degradation profiles of iron should be explored and established. In this study, we examined the degradation profile of iron only by performing histopathological analysis and micro-CT. Future studies should focus on wider aspects of the iron degradation profile, particularly under different physiological conditions. Third, we used a small animal in this study; however, the implant volume and weight were designed for human use. To obtain more accurate and realistic data, future studies should use animals with physiological parameters similar to those of humans.

## 4. Materials and Methods

### 4.1. Production and In Vitro Tests of Iron SAs Developed Using AM Technology

The innovative porous iron SA was produced using AM selective laser sintering technology (SLM EOSINT M 270 model; EOS GambH-Electro Optical Systems, Krailling, Germany). The suture anchors were designed to have a circular cross section with double- and triple-helical threads and an integrated porous structure to increase their surface area and provide high initial stability after implantation (Figure 1A). The suture anchors were prepared using the bioabsorbable spherical iron powder with an Fe purity of >99.5% [40].

In vitro mechanical tests were conducted to evaluate the mechanical characteristics of the suture anchors. The tests were performed using a 15-pound per cubic foot (pcf) polyurethane foam block (part#1522-02; Sawbone, Pacific Research Laboratories, Vashon, WA, USA). For comparison, commercialized bioabsorbable polymer SAs (5.5-mm Healicoil PK Suture Anchor, Smith & Nephew, London, UK) were used as the control. A No. 2 high-tensile-strength suture (FiberWire, Arthrex, Naples, FL, USA) with equal limbs was threaded through the suture eyelet, looped, and fixed over a post on the adapter before mechanical testing. The static ultimate pullout strength was examined at a displacement rate of 12.5 mm/s. The mechanical tests were performed using Instron E3000 (ElectroPuls, Instron, MA, USA).

The 3-(4,5-dimethylthiazol-2-yl)-2,5-diphenyl tetrazolium bromide (MTT) assay (M6494, Invitrogen, Thermo Fisher Scientific, Waltham, MA, USA) and scanning electron microscope (SEM) (DSM940 Zeiss model, Carl-Zeiss AG, Oberkochen, Germany) analyses were performed in vitro to evaluate the cytotoxicity of the suture anchor. For the MTT assay, mouse fibroblast cells (CRCC 60091 NCTN Clone 929) were used. The cells were seeded into 96-well plates (1 × 104 cells/well) and cultured in α-MEM supplemented with 10% fetal bovine serum at 37 °C in 5% CO_2_ for 24 h. Subsequently, the cells were treated with 100μL the iron SAs extract (the extraction ratio of the iron SAs was 0.2 g in 1mL α-MEM supplemented with 10% fetal bovine serum) and incubated at 37 °C in 5% CO_2_ for 24 h [40]. After the iron SAs extract treatment, the cells were incubated with MTT (50 μL/well) at 37 °C) in 5% CO_2_ for 4 h, and then 200 μL of DMSO was added. The absorbance of individual wells was determined at 570 nm. For the SEM examination, MG-63 cells (ATCC^®^ CRL-1427TM), a human bone osteosarcoma cell line, were incubated in αMEM supplemented with 10% fetal bovine serum and iron SAs (2 × 105/50 μL for each implant) at 37 °C in 5% CO_2_ for 14 days before performing SEM examination.

### 4.2. In Vivo Animal Study Design

All animal experiments were approved by the Ethics Committee of the Biomedical Technology and Device Research Laboratories of Industrial Technology Research Institute in accordance with national animal welfare legislation (approval no.: MI-20190602), and the study protocol conformed to the National Institute of Health guidelines for the use of laboratory animals. A total of 54 New Zealand white rabbits (Master Laboratory Co., Taiwan) with a mean body weight of 3.5 ± 0.5 kg at the age of 6 months were selected. The rabbits were randomized into experimental and control groups by using the computer-generated randomization method. In the control group, a polymer SA was implanted in one of the distal femoral condyles of the stifle joints. By contrast, in the experimental group, a bioabsorbable iron SA with double- or triple-helical threads was implanted using the same surgical procedure as in the control group (Iron SA_2 helix and Iron SA_3 helix groups, respectively). All the three groups were further divided into two subcategories based on the implantation periods of 4 and 12 weeks after surgery (18 in each group). Histological analysis was performed in six rabbits in each group, whereas micro-CT and biomechanical tests were performed in the remaining 12 rabbits. Micro-CT was performed in each animal immediately at the end of each experiment. Subsequently, the specimens were freshly frozen for use in further biomechanical tests.

### 4.3. Surgical Methods

All surgical procedures were performed under general anesthesia by administering an intramuscular injection of a Zoletil–Rompun mixture (Zoletil 15 mg/kg; Rompun 0.05 mL/kg; Zoletil, Virbac Taiwan, Taipei, Taiwan; Rompun, Bayer Taiwan, Taipei, Taiwan). To induce analgesia, the rabbits were given meloxicam (0.15 mg/kg peroral; Metacam, Boehringer Ingelheim Taiwan, Taiwan) 1 day preoperatively, immediately preoperatively, and 2 days following surgery.

Surgical procedures were performed following the method reported by Yamakado et al. with some modification [41]. Briefly, lateral parapatellar arthrotomy was performed in one of the stifle joints to gain access to the outer cortex of the lateral femoral condyle. A tapered hole (4.5-mm in maximal diameter) was made using a bone awl perpendicular to the long axis of the femur. Subsequently, the suture anchor was screwed in through the axis of the hole. After correct placement of the suture anchor, the joint capsule, muscles, subcutis, and cutis were separately closed using an absorbable suture material (Vicryl 4–0; Ethicon, NJ, USA). After surgery, the animals were returned to their cages and were free to move without any restriction or immobilization of their extremities. All animals were found to be ambulant without signs of guarding or immobility when they were sacrificed. For biochemical analysis, the concentrations of iron, alanine transaminase (ALT), creatinine (Cr), and blood urea nitrogen (BUN) were determined preoperatively and at 4, 8, and 12 weeks postoperatively. At the end of experiments, all animals were euthanized by administering an intravenous overdose of pentobarbital. Their liver, kidneys, heart, spleen, and stifle joints were retrieved and stored at −20 °C until future analysis.

### 4.4. Biomechanical Analysis

Six rabbits were sacrificed at 4 and 12 weeks postoperatively, respectively, and their stifle joints were retrieved and used in biomechanical analysis. The distal femur was harvested along with the implants. A material testing machine (Instron E3000; ElectroPuls, Instron, MA, USA) was used for biomechanical testing. The test was performed at room temperature (25 °C) in a moist environment. After the removal of the redundant tissue, the femur was fixed at the base plate with the long axis of the implant being parallel to the tensile force direction, and a No. 2 high-tensile-strength suture (FiberWire, Arthrex, Naples, FL, USA) with equal limbs was threaded through the suture eyelet, looped, and fixed over a post on the load cell. The tensile load parallel to the long axis of the suture anchors was examined at a strain rate of 0.5 mm/min until the occurrence of failure. The ultimate pullout load and the failure mode of the constructs were recorded and analyzed.

### 4.5. Micro-CT Analysis

After sacrificing the rabbits, six specimens were retrieved from each group and scanned using a multi-scale nano-CT (Skyscan 2211, Bruker Micro-CT, Kontich, Belgium) at a voxel resolution of 18 μm. A 360° scan with a high voltage of 160 kVp, a current of 140 μA, and an output of 20 W was conducted. Image reconstruction was performed using the reconstruction software InstaRecon xCBR (version 2.0.4.6, InstaRecon, Champaign, IL, USA) and NRecon (Bruker Micro-CT, Kontich, Belgium). Ring artifact and beam-hardening correction were performed using NRecon (Bruker Micro-CT, Kontich, Belgium).

Reconstructed cross sections were reorientated, and the region of interest (ROI) was selected. We performed the analysis using 5-mm (278 slices) images. Thresholding and bone growth analysis were performed using CTAn software. The ROI of the implant (6.25 mm in diameter) was segmented before performing bone growth analysis. A 0–1000-μm region around the implant was defined as the ROI for the bone growth analysis (Figure 4A). The metallic structure and bone were separately isolated based on the difference in X-ray absorption. The border of the metallic structure was examined using CTAn software with the shrink-wrap algorithm. Tissue volume (TV, mm^3^), bone volume (BV, mm^3^), percent bone volume (BV/TV, %), bone surface area (BS, mm^2^), and bone surface area per total volume (BS/TV, 1/mm) were measured 0–1000 μm above the metallic implant bone. Subsequently, a “sphere-fitting” measurement method was used to analyze the thickness of the structure and the implant structure (ST; mm) [42,43,44]. In addition to the bone formation analysis, implant degradation profile analysis was performed. Small implant fragments with a diameter of <0.18 mm that were detected in the ROI were defined as the small-structure volume (SSV), and the implant volume in the ROI was defined as the object volume (OV). SSV/OV (%) denoted the percentage of small fragments to that of the implant in the ROI. Three-dimensional visualization was performed using Avizo software (Thermo Fisher Scientific, Waltham, MA, USA) and CTVox (Bruker Micro-CT, Kontich, Belgium).

### 4.6. Histological Analysis

Four specimens were retrieved from each group for histological analysis 3 months postoperatively. All the harvested samples were fixed in 10% formalin for 14 days and sequentially dehydrated with increasing concentrations of ethanol (70%, 95%, and 100%) for at least 1 day and infiltrated for 5 days by using polymethylmethacrylate. After embedding, the samples were cut vertically, perpendicular to the long axis of the suture anchor, at the level of the respective bone–implant interfaces. The sections were cut to approximately 150 μm in thickness by using a low-speed saw (IsoMet, Buehler, Lake Bluff, IL, USA) and ground to 60 μm by using a grinding and polishing machine. The ground sections were stained with Sanderson’s rapid bone stain (Dorn & Hart Microedge Inc., Loxley, AL, USA) and then counterstained with acid fuchsin. All bone–implant interfaces were carefully examined under a light microscope (Nikon Eclipse Ti-series, Melville, NY, USA).

Nine sets of visceral organs, namely the liver, kidney, heart, and spleen, from each group were stained with hematoxylin and eosin (H&E), and the remaining sets were stained with Prussian blue and semi-quantified under a light microscope.

### 4.7. Statistical Analysis

All experimental data are presented as the mean ± standard deviation, with values obtained from more than three experiments. The Wilcoxon rank-sum test and Fisher’s exact test were used for nonparametric analysis. Data of more than two groups were compared using one-way analysis of variance and Tukey’s post hoc test for repeated measures. The correlation was examined by determining Pearson and Spearman correlation coefficients. A *p* value of < 0.05 was considered statistically significant. Statistical analysis was performed using PASW software (SPSS, Chicago, IL, USA).

## 5. Conclusions

In conclusion, the results of the present study exhibited that the iron SAs with triple-helical threads had the significantly better ultimate pull-out strength compared to the iron SAs with double-helical counterparts and synthetic polymer SAs, especially at the early stage after implantation. On the other hand, the open architecture design and integrated porous structure of bioabsorbable iron SAs demonstrated better biocompatibility both locally and systemically after short-term implantation. The histopathological examination of visceral organs (heart, kidney, liver, and spleen) in the iron SA groups showed no significant differences compared to those in the synthetic polymer SA group. In summary, the bioabsorbable iron SAs combined the advantages of the mechanical superiority of the iron metal and the possibility of absorption after implantation, making them a suitable candidate for further development.

## Figures and Tables

**Figure 1 ijms-22-07368-f001:**
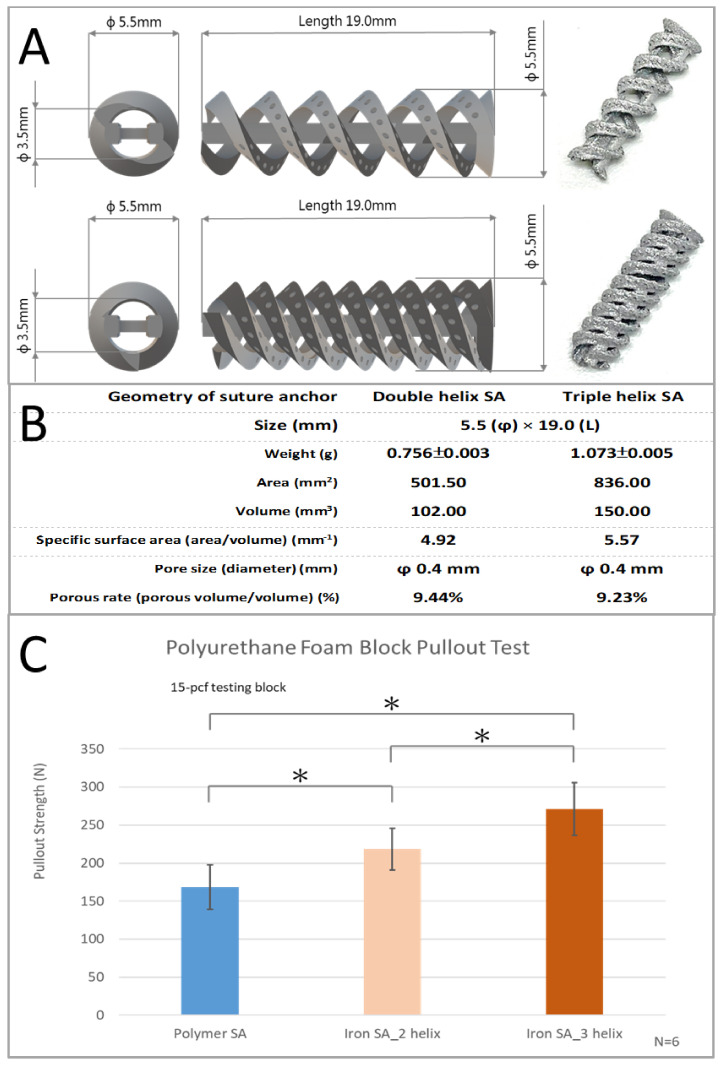
Illustration of iron-based bioabsorbable porous double- and triple-helical suture anchors (**A**). Geometrical specifications of iron-based bioabsorbable porous double- and triple-helical suture anchors (**B**). In vitro mechanical ultimate pullout strength test of three types of suture anchors (**C**). The error bar represents the standard deviation, and * denotes statistical significance between the two groups.

**Figure 2 ijms-22-07368-f002:**
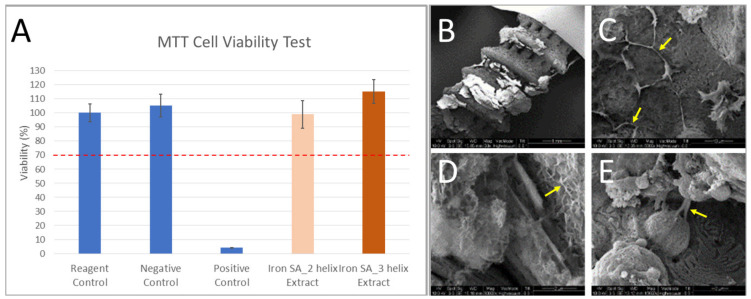
The 3-(4,5-dimethylthiazol-2-yl)-2,5-diphenyl tetrazolium bromide (MTT) assays were performed for the extracts of iron-based bioabsorbable porous suture anchors of different groups (**A**) which showed >70% cell viability. Scanning electron microscope (SEM) analysis of iron-based bioabsorbable porous testing suture anchors (prototype, not final version) (**B**) demonstrated cell attachment on the implant surface. Yellow arrows indicated the lamellipodial and filopodial extrusions from the cells (**C**–**E**). The error bar represented the standard deviation.

**Figure 3 ijms-22-07368-f003:**
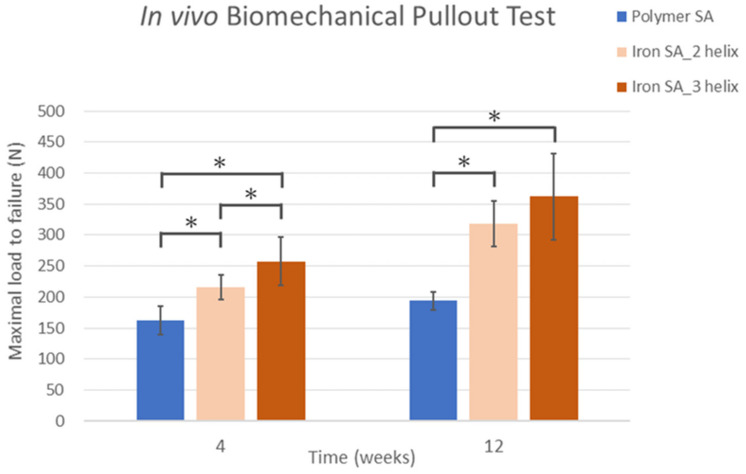
In vivo biomechanical ultimate pullout strength assessment for different suture anchors. The ultimate failure load of the iron SA_3 helix group at 4 weeks was significantly higher than those of the iron SA_2 helix and Polymer SA groups. At 12 weeks, the ultimate failure load of the iron SA_2 helix and iron SA_3 helix groups were significantly higher than that of the polymer SA group The error bar represents the standard deviation, and * denotes statistical significance between the two groups.

**Figure 4 ijms-22-07368-f004:**
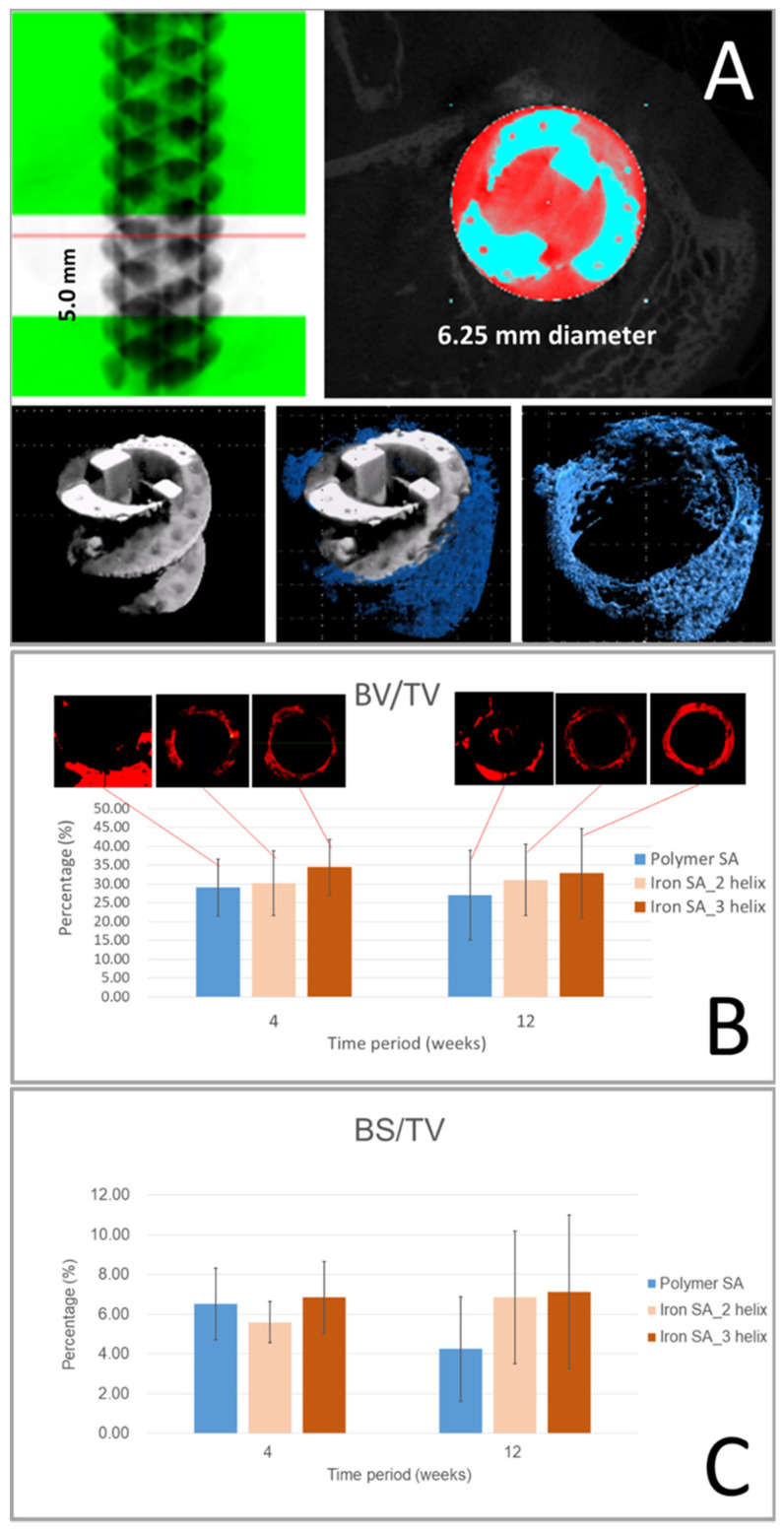
Micro-computed tomography (Micro-CT) analysis. The region of interest (ROI; diameter, 6.25 mm; thickness, 5 mm) of the bone–implant block was segmented before bone analysis, and bone growth was examined 0–1000 μm around the implant (**A**). Quantitative evaluation of bone volume between bone and suture anchors. Tissue volume (TV, mm3), bone volume (BV, mm3), and bone surface (BS, mm2) were examined 0–1000 μm above the implant surface. Bone volume fraction (BV/TV) (**B**) and bone surface density (BS/TV) (**C**) represent the bone volume rate and bone tissue surface rate, respectively. The error bar represents the standard deviation.

**Figure 5 ijms-22-07368-f005:**
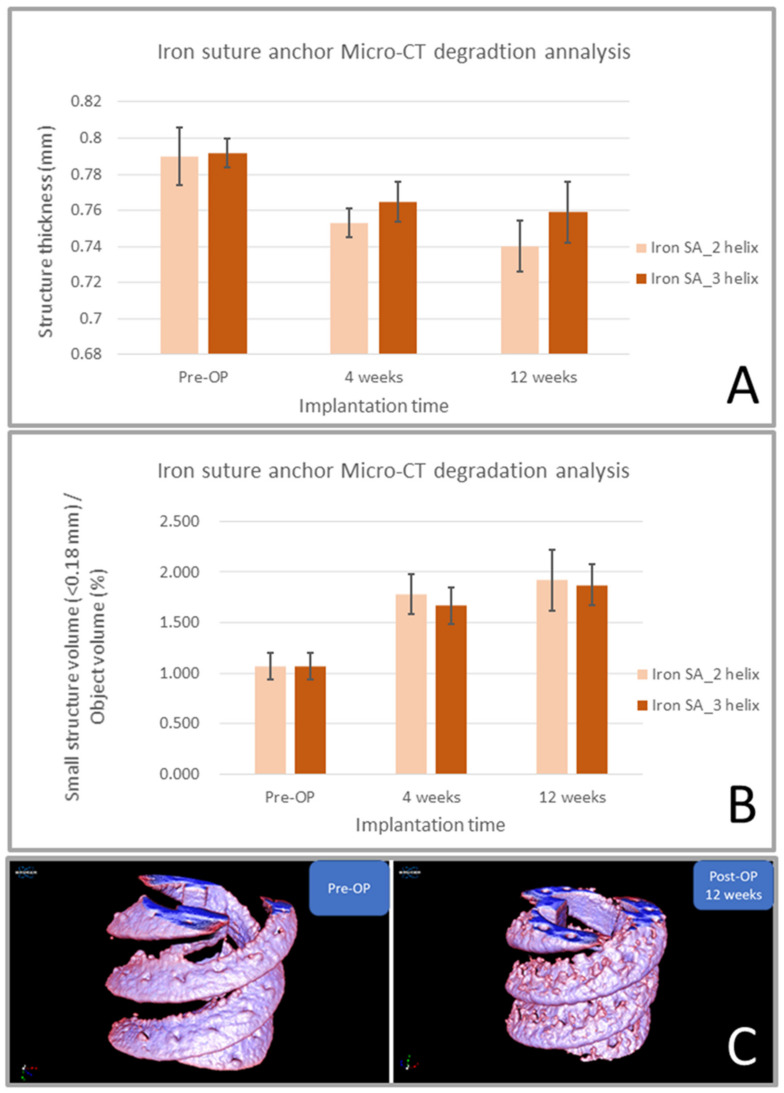
Micro-computed tomography (CT) degradation analysis of the iron suture groups. ST decreased (**A**) and SSV/OV percentage increased (**B**) sequentially from the preoperative period until 12 weeks postoperatively. Reconstructed micro-CT images observed preoperatively and at 12 weeks postoperatively (**C**). ST: structure thickness (mm), SSV: small implant fragments < 0.18 mm in diameter (small-structure volume), OV: object volume, and SSV/OV (%) denoted the small fragment percentage of the implant in the region of interest. The error bar represents the standard deviation.

**Figure 6 ijms-22-07368-f006:**
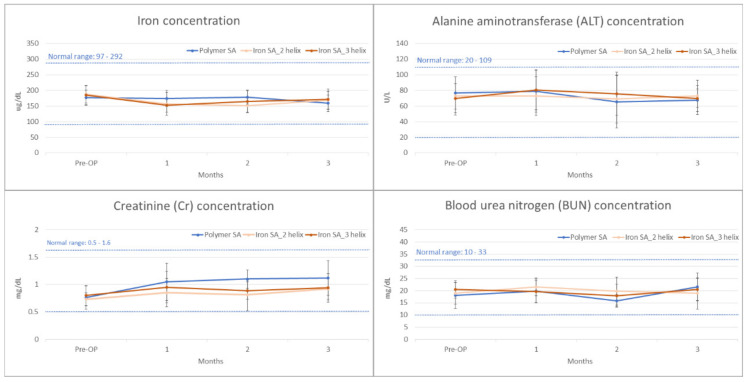
Blood concentrations of iron (μg/dL), alanine transaminase (ALT; U/L), creatinine (Cr; mg/dL), and blood urea nitrogen (BUN; mg/dL) preoperatively and 1, 2, and 3 months postoperatively. The error bar represented the standard deviation.

**Figure 7 ijms-22-07368-f007:**
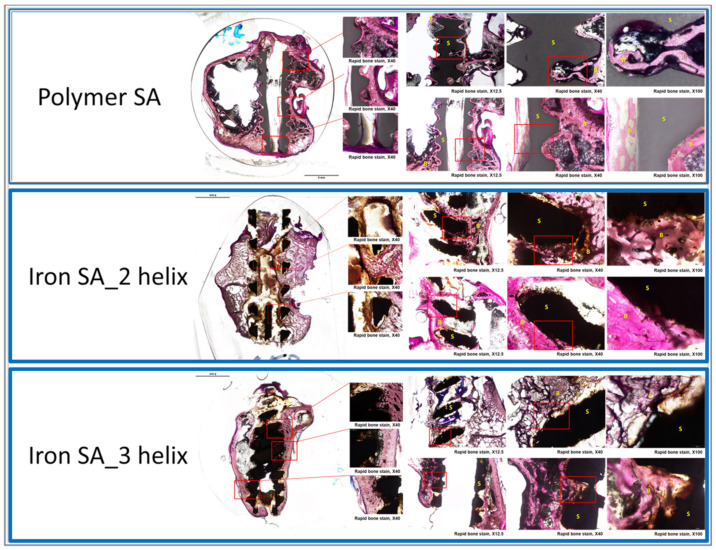
Histological examination of the bone–suture anchor interface. The specimens were stained with Sanderson’s rapid bone stain and then counterstained with acid fuchsin. Scale from 12.5×, 40×, and 100×, respectively. B: bone, S: suture anchor.

**Figure 8 ijms-22-07368-f008:**
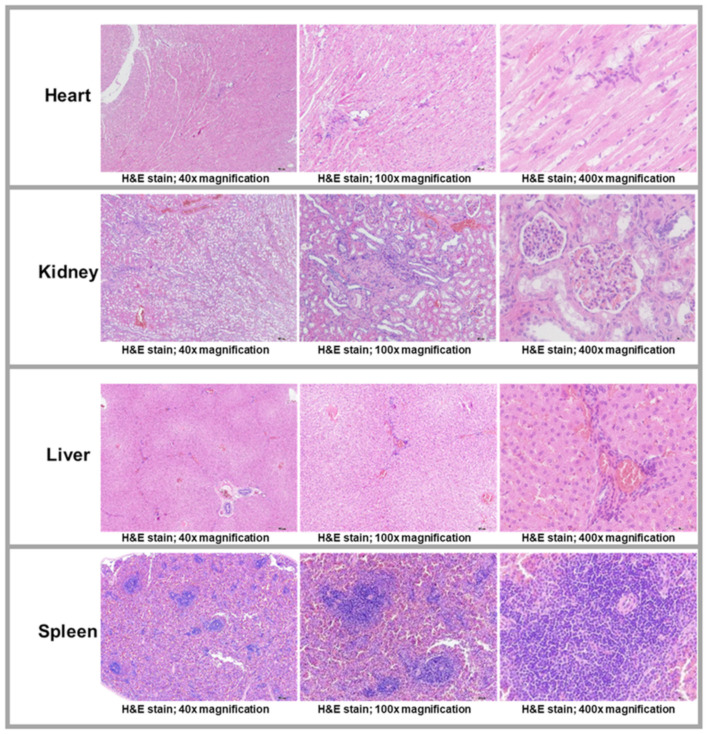
Histopathological examinations of visceral organs (heart, kidney, liver, and spleen). The specimens were obtained from one of the animals implanted with an iron bioabsorbable triple-helix suture anchor. Scale from 40×, 100×, and 400×, respectively. H&E stain: hematoxylin and eosin stain.

**Figure 9 ijms-22-07368-f009:**
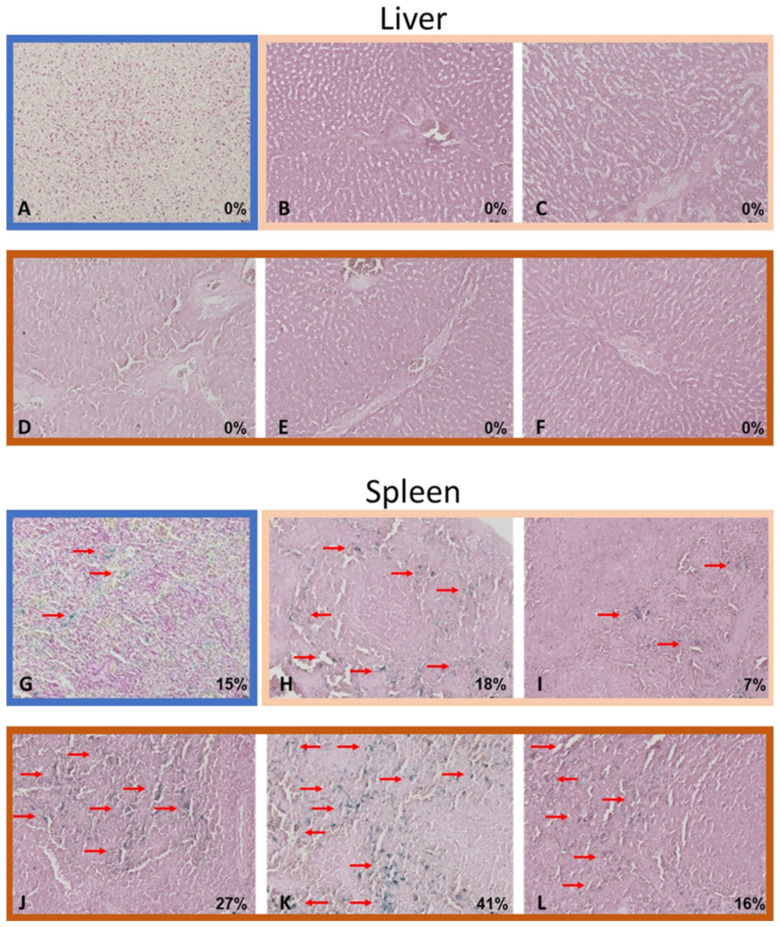
Prussian blue staining of the liver (**A**–**F**) and spleen (**G**–**L**). The specimens were obtained from the polymer SA group (**A**,**G**), iron SA_2 helix group (**B**,**C**,**H**,**I**), and iron SA_3 helix group (**D**–**F**,**J**–**L**), respectively. Red arrows indicate identified iron stain clusters. Magnification: 200×.

**Table 1 ijms-22-07368-t001:** In vitro ultimate pullout strength of suture anchors.

Group	Pullout Strength (N)	*p* Value vs. Polymer SA	*p* Value vs. Polymer SA
Polymer SA	168.54 ± 29.45		
Iron SA_2 Helix	218.43 ± 27.53	0.0126	
Iron SA_3 Helix	270.94 ± 34.76	0.0003	0.0158

**Table 2 ijms-22-07368-t002:** MTT cell viability assay.

	Viability ± SD (%)
Reagent Control	100 ± 6.46
Negative Control	105.1 ± 7.95
Positive Control	4.22 ± 0.12
Iron SA_2 Helix Extract	98.83 ± 9.61
Iron SA_3 Helix Extract	115.08 ± 8.45

**Table 3 ijms-22-07368-t003:** In vivo biomechanical ultimate pullout test.

Inter-Group Statistical Analysis				
Weeks	Group	Pullout Strength (N)	*p* Value vs. Polymer SA	*p* Value vs. Iron SA_2 Helix
4	Polymer SA	162.36 ± 23.02		
	Iron SA_2 Helix	216.23 ± 20.07	0.015	
	Iron SA_3 Helix	257.85 ± 38.66	0.004	0.0413
12	Polymer SA	193.80 ± 14.28		
	Iron SA_2 Helix	318.59 ± 36.58	0.0001	
	Iron SA_3 Helix	361.97 ± 69.45	0.0002	0.2056
Intra-group statistical analysis				
Group		*p* value between 4 vs. 12 weeks		
Polymer SA		0.0175		
Iron SA_2 helix		0.0001		
Iron SA_3 helix		0.0094		

**Table 4 ijms-22-07368-t004:** Micro-CT analysis.

BV/TV Inter-Group Statistical Analysis				
Weeks	Group	BV/TV (%)	*p* Value vs. Polymer SA	*p* Value vs. Iron SA_2 Helix
4	Polymer SA	29.04 ± 7.57		
	Iron SA_2 Helix	30.19 ± 8.55	0.8102	
	Iron SA_3 Helix	34.42 ± 7.33	0.2395	0.3792
12	Polymer SA	27.01 ± 11.86		
	Iron SA_2 Helix	31.02 ± 9.46	0.5319	
	Iron SA_3 Helix	32.89 ± 11.81	0.4097	0.7683
BV/TV Intra-Group Statistical Analysis				
Group		*p* Value between 4 vs. 12 Weeks		
Polymer SA		0.7311		
Iron SA_2 Helix		0.8765		
Iron SA_3 Helix		0.7929		
BS/TV Inter-Group Statistical Analysis				
Weeks	Group	BS/TV (%)	*p* Value vs. Polymer SA	*p* Value vs. Iron SA_2 Helix
4	Polymer SA	6.52 ± 1.80		
	Iron SA_2 Helix	5.60 ± 1.04	0.3038	
	Iron SA_3 Helix	6.84 ± 1.82	0.7657	0.1780
12	Polymer SA	4.25 ± 2.64		
	Iron SA_2 Helix	6.85 ± 3.34	0.1656	
	Iron SA_3 Helix	7.13 ± 3.87	0.1630	0.8959
BS/TV Intra-Group Statistical Analysis				
Group		*p* value between 4 vs. 12 Weeks		
Polymer SA		0.1124		
Iron SA_2 Helix		0.4020		
Iron SA_3 Helix		0.8714		

**Table 5 ijms-22-07368-t005:** Micro-CT degradation analysis of the iron suture anchor for structure thickness (ST) and small-structure volume (<0.18 mm) (SSV)/object volume (OV).

**ST Inter-Group Statistical Analysis**				
**Weeks**	**Group**	**ST (mm)**	***p* Value vs. Pre-OP**	***p* Value vs. 4 Weeks**
Pre-OP	Iron SA_2 helix	0.790 ± 0.016		
	Iron SA_3 helix	0.792 ± 0.008		
4	Iron SA_2 helix	0.753 ± 0.008	0.0005	
	Iron SA_3 helix	0.765 ± 0.011	0.0007	
12	Iron SA_2 helix	0.740 ± 0.014	0.0002	0.0765
	Iron SA_3 helix	0.759 ± 0.017	0.0016	0.4846
ST Intra-Group Statistical Analysis				
Time period		*p* value between 2 vs. 3 helix		
Pre-OP		0.7898		
4 weeks		0.056		
12 weeks		0.0607		
**SSV/OV Inter-Group Statistical Analysis**				
**Weeks**	**Group**	**SSV/OV (%)**	***p* Value vs. Pre-OP**	***p* Value vs. 4 Weeks**
Pre-OP	Iron SA_2 helix	1.068 ± 0.128		
	Iron SA_3 helix	1.064 ± 0.132		
4	Iron SA_2 helix	1.780 ± 0.194	0.0001	
	Iron SA_3 helix	1.666 ± 0.179	0.0001	
12	Iron SA_2 helix	1.920 ± 0.301	0.0001	0.3608
	Iron SA_3 helix	1.871 ± 0.205	0.0001	0.0948
SSV/OV Intra-Group Statistical Analysis				
Time period		*p* value between 2 vs. 3 Helix		
Pre-OP		0.9586		
4 weeks		0.315		
12 weeks		0.7485		

**Table 6 ijms-22-07368-t006:** Blood biochemistry analysis.

	Iron Concentration (μg/dL)			
	Pre-OP	4 Weeks	8 Weeks	12 Weeks
Polymer SA	176.39 ± 24.84	173.51 ± 25.49	178.48 ± 21.97	158.43 ± 26.28
Iron SA_2 Helix	186.85 ± 28.54	156.87 ± 36.52	150.87 ± 23.68	168.96 ± 28.57
Iron SA_3 Helix	184.54 ± 30.58	152.47 ± 22.25	164.86 ± 34.85	170.65 ± 32.58
	Alanine Transaminase (ALT) Concentration (U/L)			
	Pre-OP	4 weeks	8 weeks	12 weeks
Polymer SA	76.58 ± 20.58	78.52 ± 26.54	65.44 ± 33.52	67.61 ± 18.72
Iron SA_2 Helix	72.86 ± 24.27	72.54 ± 24.20	68.84 ± 30.79	72.86 ± 20.11
Iron SA_3 Helix	69.68 ± 18.57	80.54 ± 25.67	75.28 ± 27.65	69.81 ± 11.92
	Creatinine (Cr) Concentration (mg/dL)			
	Pre-OP	4 weeks	8 weeks	12 weeks
Polymer SA	0.76 ± 0.21	1.05 ± 0.34	1.10 ± 0.17	1.12 ± 0.32
Iron SA_2 Helix	0.73 ± 0.12	0.85 ± 0.26	0.81 ± 0.29	0.92 ± 0.20
Iron SA_3 Helix	0.80 ± 0.18	0.95 ± 0.29	0.89 ± 0.16	0.94 ± 0.26
	Blood Urea Nitrogen (BUN) Concentration (mg/dL)			
	Pre-OP	4 weeks	8 weeks	12 weeks
Polymer SA	17.99 ± 5.31	19.85 ± 4.97	15.87 ± 2.59	21.52 ± 5.81
Iron SA_2 Helix	18.85 ± 4.52	21.54 ± 3.58	19.83 ± 5.78	18.85 ± 6.58
Iron SA_3 Helix	20.54 ± 3.67	19.63 ± 4.57	17.88 ± 4.73	20.52 ± 4.52

**Table 7 ijms-22-07368-t007:** Semi-quantitative results for the Prussian blue staining of the liver and spleen.

Semi-Quantitative Analysis for Prussian-Blue (Iron Stain) of Spleen				
	Simple Scoring of Spleen Iron Store	Percentage of Spleen Iron Store (%)	*p* Value vs. Polymer SA	*p* Value vs. Iron SA_2 Helix
Polymer SA	+	25.5 ± 14.8		
Iron SA_2 Helix	+	22.4 ± 11.8	0.6967	
Iron SA_3 Helix	+	24.9 ± 16.1	0.9477	0.7653
Semi-Quantitative Analysis for Prussian-Blue (Iron Stain) of Liver				
	Simple Scoring for Iron Store in Liver			
Polymer SA	Not Detectable (ND)			
Iron SA_2 Helix	Not Detectable (ND)			
Iron SA_3 Helix	Not Detectable (ND)			

## Data Availability

The datasets generated and/or analyzed during the current study are not publicly available because they contain trade secrets but can be made available from the corresponding author on reasonable request.

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
