# Peer review of "Biocompatibility and Biological Performance Evaluation of Additive-Manufactured Bioabsorbable Iron-Based Porous Suture Anchor in a Rabbit Model"

_ijms, 2021, doi:10.3390/ijms22147368_

Round 1
Reviewer 1 Report
The manuscript presented by Tai et al. brings the compelling study of AM biodegradable iron sutures with novel structures examined in vivo. A complex biological evaluation of the iron-based samples along with the study of their mechanical properties was conducted using a polymer sample as a reference. Various biological methods (CT scans, histological analysis, SEM analysis, MTT assay) were used to characterize the biocompatibility of the sutures. This study may be of great interest to readers in the field of degradable metals. However, several issues need to be addressed before publication:
Q1: More references supporting statements in the introduction should be added.
Q2: Motivation for choice and comparison with other studies concerning the unique helical structure of the proposed material should be also added to the introduction. The authors should emphasize why was this geometry used. Was the initial weight of the iron samples the same for both types?
Q3: Chapter 2.1 should be divided into 2 separate sections for mechanical and biological analysis.
Q4: What was the pore size diameter for prepared samples and did it differ for -2 vs. -3 helix samples?
Q5: Was MTT assay performed on the polymer reference sample?
Q6: In Figure 2B, which sample is visualized? Figure 2b should be divided and marked with 4 different letters.
Q7: What do the authors mean by using the term "biomechanical"?
Q8: In Figure 3, the results from the Pullout test are presented after several weeks. How would the authors explain that the values after 12 weeks are higher than that after 4 weeks? This should be discussed.
Q9: In Table 6, iron concentration during the period of 12 weeks was monitored. How would the authors explain that the levels of iron decreased after 4 weeks where the most rapid degradation was recorded? Moreover, these values were higher for the polymer sample after 4 and 8 weeks.
Q10: Texts in Figures 6 and 7 are not clearly visible.
Q11: Were the histological analysis performed on the polymer sample (Figure 9)?
Q12: How would the authors explain no significant increase of iron?
Q13: The discussion part of the paper summarizes the findings observed in the study, however, a detailed scientific explanation for observed findings should be added considering sample geometry differences e.g.
Q14: Conclusions should be slightly widened.

Author Response
Thank you very much for your kind suggestions about this manuscript. We appreciate your good point and already made some revisions for your kind reminders. Thank you for helping us polish our manuscript. Please find the attached file for the list of responses to reviewers.

Reviewer 2 Report
Please check English language throughout the manuscript. Otherwise, I don't have additional comments.
Author Response
Thank you very much for your kind suggestions about this manuscript. We appreciate your good point and already made some revisions for your kind reminders. Thank you for helping us polish our manuscript. We have already revised the English throughout the manuscript.